# Analysis of Learning from
# Positive and Unlabeled Data

**Marthinus C. du Plessis**
The University of Tokyo
Tokyo, 113-0033, Japan
christo@ms.k.u-tokyo.ac.jp

**Gang Niu**
Baidu Inc.
Beijing, 100085, China
niugang@baidu.com

**Masashi Sugiyama**
The University of Tokyo
Tokyo, 113-0033, Japan
sugi@k.u-tokyo.ac.jp

## Abstract

*Learning a classifier from positive and unlabeled data* is an important class of classification problems that are conceivable in many practical applications. In this paper, we first show that this problem can be solved by cost-sensitive learning between positive and unlabeled data. We then show that convex surrogate loss functions such as the hinge loss may lead to a wrong classification boundary due to an intrinsic bias, but the problem can be avoided by using non-convex loss functions such as the ramp loss. We next analyze the excess risk when the class prior is estimated from data, and show that the classification accuracy is not sensitive to class prior estimation if the unlabeled data is dominated by the positive data (this is naturally satisfied in inlier-based outlier detection because inliers are dominant in the unlabeled dataset). Finally, we provide generalization error bounds and show that, for an equal number of labeled and unlabeled samples, the generalization error of learning only from positive and unlabeled samples is no worse than $2\sqrt{2}$ times the fully supervised case. These theoretical findings are also validated through experiments.

## 1 Introduction

Let us consider the problem of *learning a classifier from positive and unlabeled data* (PU classification), which is aimed at assigning labels to the unlabeled dataset [1]. PU classification is conceivable in various applications such as land-cover classification [2], where positive samples (built-up urban areas) can be easily obtained, but negative samples (rural areas) are too diverse to be labeled. Outlier detection in unlabeled data based on inlier data can also be regarded as PU classification [3, 4].

In this paper, we first explain that, if the class prior in the unlabeled dataset is known, PU classification can be reduced to the problem of *cost-sensitive classification* [5] between positive and unlabeled data. Thus, in principle, the PU classification problem can be solved by a standard cost-sensitive classifier such as the weighted support vector machine [6]. The goal of this paper is to give new insight into this PU classification algorithm. Our contributions are three folds:

- The use of convex surrogate loss functions such as the *hinge loss* may potentially lead to a wrong classification boundary being selected, even when the underlying classes are completely separable. To obtain the correct classification boundary, the use of non-convex loss functions such as the *ramp loss* is essential.

- When the class prior in the unlabeled dataset is estimated from data, the classification error is governed by what we call the *effective class prior* that depends both on the true class prior and the estimated class prior. In addition to gaining intuition behind the classification error incurred in PU classification, a practical outcome of this analysis is that the classification error is not sensitive to class-prior estimation error if the unlabeled data is dominated by positive data. This would be useful in, e.g., inlier-based outlier detection scenarios where inlier samples are dominant in the unlabeled dataset [3, 4]. This analysis can be regarded as an extension of traditional analysis of class priors in ordinary classification scenarios [7, 8] to PU classification.

- We establish generalization error bounds for PU classification. For an equal number of positive and unlabeled samples, the convergence rate is no worse than $2\sqrt{2}$ times the fully supervised case.

Finally, we numerically illustrate the above theoretical findings through experiments.

## 2 PU classification as cost-sensitive classification

In this section, we show that the problem of PU classification can be cast as cost-sensitive classification.

**Ordinary classification:** The Bayes optimal classifier corresponds to the decision function $f(X) \in \{1, -1\}$ that minimizes the *expected misclassification rate* w.r.t. a class prior of $\pi$:

$$R(f) := \pi R_1(f) + (1 - \pi)R_{-1}(f),$$

where $R_{-1}(f)$ and $R_1(f)$ denote the expected *false positive rate* and expected *false negative rate*:

$$R_{-1}(f) = P_{-1}(f(X) \neq -1) \quad \text{and} \quad R_1(f) = P_1(f(X) \neq 1),$$

and $P_1$ and $P_{-1}$ denote the marginal probabilities of positive and negative samples.

In the empirical risk minimization framework, the above risk is replaced with their empirical versions obtained from fully labeled data, leading to practical classifiers [9].

**Cost-sensitive classification:** A cost-sensitive classifier selects a function $f(X) \in \{1, -1\}$ in order to minimize the weighted expected misclassification rate:

$$R(f) := \pi c_1 R_1(f) + (1 - \pi)c_{-1}R_{-1}(f), \tag{1}$$

where $c_1$ and $c_{-1}$ are the per-class costs [5].

Since scaling does not matter in (1), it is often useful to interpret the per-class costs as reweighting the problem according to new class priors proportional to $\pi c_1$ and $(1 - \pi)c_{-1}$.

**PU classification:** In PU classification, a classifier is learned using labeled data drawn from the positive class $P_1$ and unlabeled data that is a mixture of positive and negative samples with unknown class prior $\pi$:

$$P_X = \pi P_1 + (1 - \pi)P_{-1}.$$

Since negative samples are not available, let us train a classifier to minimize the expected misclassification rate between positive and unlabeled samples. Since we do not have negative samples in the PU classification setup, we cannot directly estimate $R_{-1}(f)$ and thus we rewrite the risk $R(f)$ not to include $R_{-1}(f)$. More specifically, let $R_X(f)$ be the probability that the function $f(X)$ gives the positive label over $P_X$ [10]:

$$\begin{aligned} R_X(f) &= P_X(f(X) = 1) \\ &= \pi P_1(f(X) = 1) + (1 - \pi)P_{-1}(f(X) = 1) \\ &= \pi(1 - R_1(f)) + (1 - \pi)R_{-1}(f). \end{aligned} \tag{2}$$

Then the risk $R(f)$ can be written as

$$\begin{aligned} R(f) &= \pi R_1(f) + (1 - \pi)R_{-1}(f) \\ &= \pi R_1(f) - \pi(1 - R_1(f)) + R_X(f) \\ &= 2\pi R_1(f) + R_X(f) - \pi. \end{aligned} \qquad (3)$$

Let $\eta$ be the proportion of samples from $P_1$ compared to $P_X$, which is empirically estimated by $\frac{n}{n+n'}$ where $n$ and $n'$ denote the numbers of positive and unlabeled samples, respectively. The risk $R(f)$ can then be expressed as

$$R(f) = c_1 \eta R_1(f) + c_X(1 - \eta)R_X(f) - \pi, \quad \text{where} \quad c_1 = \frac{2\pi}{\eta} \quad \text{and} \quad c_X = \frac{1}{1 - \eta}.$$

Comparing this expression with (1), we can confirm that the PU classification problem is solved by cost-sensitive classification between positive and unlabeled data with costs $c_1$ and $c_X$. Some implementations of support vector machines, such as `libsvm` [6], allow for assigning weights to classes. In practice, the unknown class prior $\pi$ may be estimated by the methods proposed in [10, 1, 11].

In the following sections, we analyze this algorithm.

## 3  Necessity of non-convex loss functions in PU classification

In this section, we show that solving the PU classification problem with a convex loss function may lead to a biased solution, and the use of a non-convex loss function is essential to avoid this problem.

**Loss functions in ordinary classification:**  We first consider ordinary classification problems where samples from both classes are available. Instead of a binary decision function $f(X) \in \{-1, 1\}$, a continuous decision function $g(X) \in \mathbb{R}$ such that $\text{sign}(g(X)) = f(X)$ is learned. The loss function then becomes

$$J_{0\text{-}1}(g) = \pi \mathbb{E}_1 \left[ \ell_{0\text{-}1}(g(X)) \right] + (1 - \pi)\mathbb{E}_{-1} \left[ \ell_{0\text{-}1}(-g(X)) \right],$$

where $\mathbb{E}_y$ is the expectation over $P_y$ and $\ell_{0\text{-}1}(z)$ is the zero-one loss:

$$\ell_{0\text{-}1}(z) = \begin{cases} 0 & z > 0, \\ 1 & z \leq 0. \end{cases}$$

Since the zero-one loss is hard to optimize in practice due to its discontinuous nature, it may be replaced with a *ramp loss* (as illustrated in Figure 1):

$$\ell_{\text{R}}(z) = \frac{1}{2} \max(0, \min(2, 1 - z)),$$

giving an objective function of

$$J_{\text{R}}(g) = \pi \mathbb{E}_1 \left[ \ell_{\text{R}}(g(X)) \right] + (1 - \pi)\mathbb{E}_{-1} \left[ \ell_{\text{R}}(-g(X)) \right]. \qquad (4)$$

To avoid the non-convexity of the ramp loss, the *hinge loss* is often preferred in practice:

$$\ell_{\text{H}}(z) = \frac{1}{2} \max(1 - z, 0),$$

giving an objective of

$$J_{\text{H}}(g) = \pi \mathbb{E}_1 \left[ \ell_{\text{H}}(g(X)) \right] + (1 - \pi)\mathbb{E}_{-1} \left[ \ell_{\text{H}}(-g(X)) \right]. \qquad (5)$$

One practical motivation to use the convex hinge loss instead of the non-convex ramp loss is that separability (i.e., $\min_g J_{\text{R}}(g) = 0$) implies $\ell_{\text{R}}(z) = 0$ everywhere, and for all values of $z$ for which $\ell_{\text{R}}(z) = 0$, we have $\ell_{\text{H}}(z) = 0$. Therefore, the convex hinge loss will give the same decision boundary as the non-convex ramp loss in the ordinary classification setup, under the assumption that the positive and negative samples are non-overlapping.

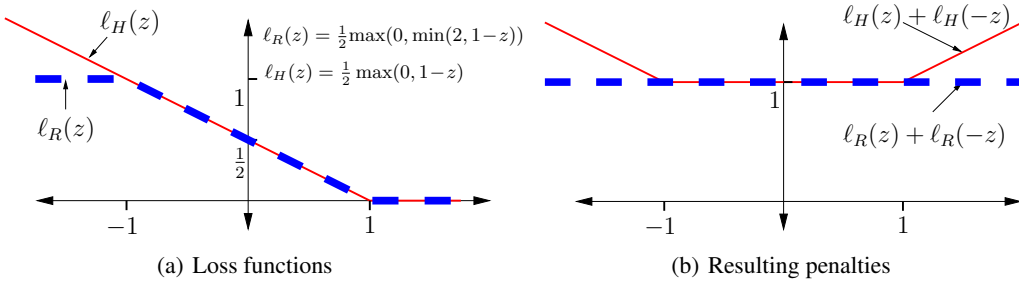

<div align="center">(a) Loss functions         (b) Resulting penalties</div>

Figure 1: $\ell_R(z)$ denotes the *ramp loss*, and $\ell_H(z)$ denotes the *hinge loss*. $\ell_R(z) + \ell_R(-z)$ is constant but $\ell_H(z) + \ell_H(-z)$ is not and therefore causes a superfluous penalty.

**Ramp loss function in PU classification:** An important question is whether the same interpretation will hold for PU classification: can the PU classification problem be solved by using the convex hinge loss? As we show below, the answer to this question is unfortunately "no".

In PU classification, the risk is given by (3), and its ramp-loss version is given by

$$
\begin{aligned}
J_{\text{PU-R}}(g) &= 2\pi R_1(f) + R_X(f) - \pi && (6) \\
&= 2\pi \mathbb{E}_1\left[\ell_R(g(X))\right] + \left[\pi \mathbb{E}_1\left[\ell_R(-g(X))\right] + (1-\pi)\mathbb{E}_{-1}\left[\ell_R(-g(X))\right]\right] - \pi && (7) \\
&= \pi \mathbb{E}_1\left[\ell_R(g(X))\right] + \pi \mathbb{E}_1\left[\ell_R(g(X)) + \ell_R(-g(X))\right] \\
&\quad + (1-\pi)\mathbb{E}_{-1}\left[\ell_R(-g(X))\right] - \pi, && (8)
\end{aligned}
$$

where (6) comes from (3) and (7) is due to the substitution of (2). Since the ramp loss is symmetric in the sense of

$$\ell_R(-z) + \ell_R(z) = 1,$$

(8) yields

$$J_{\text{PU-R}}(g) = \pi \mathbb{E}_1\left[\ell_R(g(X))\right] + (1-\pi)\mathbb{E}_{-1}\left[\ell_R(-g(X))\right]. \qquad (9)$$

(9) is essentially the same as (4), meaning that learning with the ramp loss in the PU classification setting will give the same classification boundary as in the ordinary classification setting.

For non-convex optimization with the ramp loss, see [12, 13].

**Hinge loss function in PU classification:** On the other hand, using the hinge loss to minimize (3) for PU learning gives

$$
\begin{aligned}
J_{\text{PU-H}}(g) &= 2\pi \mathbb{E}_1\left[\ell_H(g(X))\right] + \left[\pi \mathbb{E}_1\left[\ell_H(-g(X))\right] + (1-\pi)\mathbb{E}_{-1}\left[\ell_H(-g(X))\right]\right] - \pi, && (10) \\
&= \underbrace{\pi \mathbb{E}_1\left[\ell_H\left(g(X)\right)\right] + (1-\pi)\mathbb{E}_{-1}\left[\ell_H(-g(X))\right]}_{\text{Ordinary error term, cf. (5)}} + \underbrace{\pi \mathbb{E}_1\left[\ell_H(g(X)) + \ell_H(-g(X))\right]}_{\text{Superfluous penalty}} - \pi.
\end{aligned}
$$

We see that the hinge loss has a term that corresponds to (5), but it also has a superfluous penalty term (see also Figure 1). This penalty term may cause an incorrect classification boundary to be selected. Indeed, even if $g(X)$ perfectly separates the data, it may not minimize $J_{\text{PU-H}}(g)$ due to the superfluous penalty. To obtain the correct decision boundary, the loss function should be symmetric (and therefore non-convex). Alternatively, since the superfluous penalty term can be evaluated, it can be subtracted from the objective function. Note that, for the problem of label noise, an identical symmetry condition has been obtained [14].

**Illustration:** We illustrate the failure of the hinge loss on a toy PU classification problem with class conditional densities of:

$$p(\boldsymbol{x}|y = 1) = \mathcal{N}\left(-3, 1^2\right) \quad \text{and} \quad p(\boldsymbol{x}|y = 1) = \mathcal{N}\left(3, 1^2\right),$$

where $\mathcal{N}(\mu, \sigma^2)$ is a normal distribution with mean $\mu$ and variance $\sigma^2$. The hinge-loss objective function for PU classification, $J_{\text{PU-H}}(g)$, is minimized with a model of $g(x) = wx + b$ (the expectations in the objective function is computed via numerical integration). The optimal decision

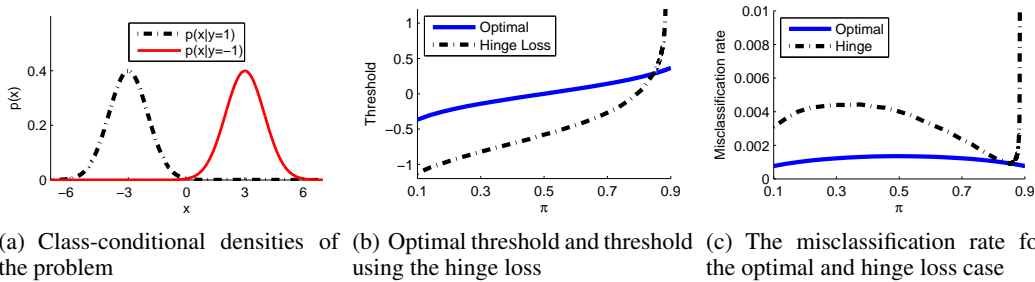

(a) Class-conditional densities of the problem

(b) Optimal threshold and threshold using the hinge loss

(c) The misclassification rate for the optimal and hinge loss case

Figure 2: Illustration of the failure of the hinge loss for PU classification. The optimal threshold and the threshold estimated by the hinge loss differ significantly (Figure 2(b)), causing a difference in the misclassification rates (Figure 2(c)). The threshold for the ramp loss agrees with the optimal threshold.

threshold and the threshold for the hinge loss is plotted in Figure 2(b) for a range of class priors. Note that the threshold for the ramp loss will correspond to the optimal threshold. From this figure, we note that the hinge-loss threshold differs from the optimal threshold. The difference is especially severe for larger class priors, due to the fact that the superfluous penalty is weighted by the class prior. When the class-prior is large enough, the large hinge-loss threshold causes all samples to be positively labeled. In such a case, the false negative rate is $R_1 = 0$ but the false positive rate is $R_{-1} = 1$. Therefore, the overall misclassification rate for the hinge loss will be $1 - \pi$.

## 4 Effect of inaccurate class-prior estimation

To solve the PU classification problem by cost-sensitive learning described in Section 2, the true class prior $\pi$ is needed. However, since it is often unknown in practice, it needs to be estimated, e.g., by the methods proposed in [10, 1, 11]. Since many of the estimation methods are biased [1, 11], it is important to understand the influence of inaccurate class-prior estimation on the classification performance. In this section, we elucidate how the error in the estimated class prior $\widehat{\pi}$ affects the classification accuracy in the PU classification setting.

**Risk with true class prior in ordinary classification:** In the ordinary classification scenarios with positive and negative samples, the risk for a classifier $f$ on a dataset with class prior $\pi$ is given as follows ([8, pp. 26–29] and [7]):

$$R(f, \pi) = \pi R_1(f) + (1 - \pi)R_{-1}(f).$$

The risk for the optimal classifier according to the class prior $\pi$ is therefore,

$$R^*(\pi) = \min_{f \in \mathcal{F}} R(f, \pi)$$

Note that $R^*(\pi)$ is *concave*, since it is the minimum of a set of functions that are linear w.r.t. $\pi$. This is illustrated in Figure 3(a).

**Excess risk with class prior estimation in ordinary classification:** Suppose we have a classifier $\widehat{f}$ that minimizes the risk for an estimated class prior $\widehat{\pi}$:

$$\widehat{f} := \arg\min_{f \in \mathcal{F}} R(f, \widehat{\pi}).$$

The risk when applying the classifier $\widehat{f}$ on a dataset with true class prior $\pi$ is then on the line tangent to the concave function $R^*(\pi)$ at $\pi = \widehat{\pi}$, as illustrated in Figure 3(a):

$$\widehat{R}(\pi) = \pi R_1(\widehat{f}) + (1 - \pi)R_{-1}(\widehat{f}).$$

The function $\widehat{f}$ is suboptimal at $\pi$, and results in the excess risk [8]:

$$E_\pi = \widehat{R}(\pi) - R(\pi).$$

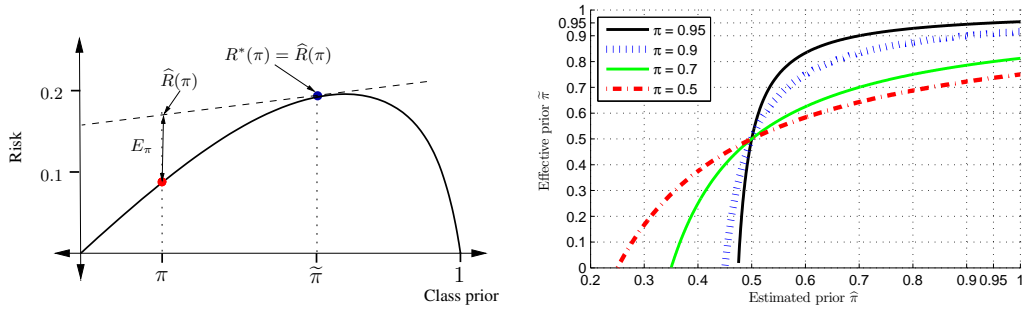

(a) Selecting a classifier to minimize (11) and apply-ing it to a dataset with class prior $\pi$ leads to an excess risk of $E_\pi$.

(b) The effective class prior $\widetilde{\pi}$ vs. the estimated class prior $\widehat{\pi}$ for different true class priors $\pi$.

Figure 3: Learning in the PU framework with an estimated class prior $\widehat{\pi}$ is equivalent to selecting a classifier which minimizes the risk according to an effective class prior $\widetilde{\pi}$. (a) The difference between the effective class prior $\widetilde{\pi}$ and the true class prior $\pi$ causes an excess risk $E_\pi$. (b) The effective class prior $\widetilde{\pi}$ depends on the true class prior $\pi$ and the estimated class prior $\widehat{\pi}$.

**Excess risk with class prior estimation in PU classification:**  We wish to select a classifier that minimizes the risk in (3). In practice, however, we only know an estimated class prior $\widehat{\pi}$. Therefore, a classifier is selected to minimize

$$R(f) = 2\widehat{\pi}R_1(f) + R_X(f) - \widehat{\pi}. \tag{11}$$

Expanding the above risk based on (2) gives

$$R(f) = 2\widehat{\pi}R_1(f) + \pi(1 - R_1(f)) + (1 - \pi)R_{-1}(f) - \widehat{\pi}$$
$$= (2\widehat{\pi} - \pi)\, R_1(f) + (1 - \pi)R_{-1}(f) + \pi - \widehat{\pi}.$$

Thus, the estimated class prior affects the risk with respect to $2\widehat{\pi} - \pi$ and $1 - \pi$. This result immediately shows that PU classification cannot be performed when the estimated class prior is less than half of the true class prior: $\widehat{\pi} \le \frac{1}{2}\pi$.

We define the *effective class prior* $\widetilde{\pi}$ so that $2\widehat{\pi} - \pi$ and $1 - \pi$ are normalized to sum to one:

$$\widetilde{\pi} = \frac{2\widehat{\pi} - \pi}{2\widehat{\pi} - \pi + 1 - \pi} = \frac{2\widehat{\pi} - \pi}{2\widehat{\pi} - 2\pi + 1}.$$

Figure 3(b) shows the profile of the effective class prior $\widetilde{\pi}$ for different $\pi$. The graph shows that when the true class prior $\pi$ is large, $\widetilde{\pi}$ tends to be flat around $\pi$. When the true class prior is known to be large (such as the proportion of inliers in inlier-based outlier detection), a rough class-prior estimator is sufficient to have a good classification performance. On the other hand, if the true class prior is small, PU classification tends to be hard and an accurate class-prior estimator is necessary.

We also see that when the true class prior is large, overestimation of the class prior is more attenu-ated. This may explain why some class-prior estimation methods [1, 11] still give a good practical performance in spite of having a positive bias.

## 5 Generalization error bounds for PU classification

In this section, we analyze the generalization error for PU classification, when training samples are clearly *not identically distributed*.

More specifically, we derive error bounds for the classification function $f(\boldsymbol{x})$ of form

$$f(\boldsymbol{x}) = \sum_{i=1}^{n} \alpha_i k(\boldsymbol{x}_i, \boldsymbol{x}) + \sum_{j=1}^{n'} \alpha_j' k(\boldsymbol{x}_j', \boldsymbol{x}),$$

where $\boldsymbol{x}_1, \ldots, \boldsymbol{x}_n$ are positive training data and $\boldsymbol{x}_1', \ldots, \boldsymbol{x}_{n'}'$ are positive and negative test data. Let

$$\mathcal{A} = \{(\alpha_1, \ldots, \alpha_n, \alpha_1', \ldots, \alpha_{n'}') \mid \boldsymbol{x}_1, \ldots, \boldsymbol{x}_n \sim p(\boldsymbol{x} \mid y = +1), \boldsymbol{x}_1', \ldots, \boldsymbol{x}_{n'}' \sim p(\boldsymbol{x})\}$$

be the set of all possible optimal solutions returned by the algorithm given some training data and test data according to $p(\boldsymbol{x} \mid y = +1)$ and $p(\boldsymbol{x})$. Then define the constants

$$C_\alpha = \sup_{\boldsymbol{\alpha} \in \mathcal{A}, \boldsymbol{x}_1, \ldots, \boldsymbol{x}_n \sim p(\boldsymbol{x}|y=+1), \boldsymbol{x}'_1, \ldots, \boldsymbol{x}'_{n'} \sim p(\boldsymbol{x})}$$

$$\left( \sum_{i,i'=1}^{n} \alpha_i \alpha_{i'} k(\boldsymbol{x}_i, \boldsymbol{x}_{i'}) + 2 \sum_{i=1}^{n} \sum_{j=1}^{n'} \alpha_i \alpha'_j k(\boldsymbol{x}_i, \boldsymbol{x}'_j) + \sum_{j,j'=1}^{n'} \alpha'_j \alpha'_{j'} k(\boldsymbol{x}'_j, \boldsymbol{x}'_{j'}) \right)^{1/2},$$

$$C_k = \sup_{\boldsymbol{x} \in \mathbb{R}^d} \sqrt{k(\boldsymbol{x}, \boldsymbol{x})},$$

and define the function class

$$\mathcal{F} = \{f : \boldsymbol{x} \mapsto \sum_{i=1}^{n} \alpha_i k(\boldsymbol{x}_i, \boldsymbol{x}) + \sum_{j=1}^{n'} \alpha'_j k(\boldsymbol{x}'_j, \boldsymbol{x}) \mid \boldsymbol{\alpha} \in \mathcal{A}, \tag{12}$$

$$\boldsymbol{x}_1, \ldots, \boldsymbol{x}_n \sim p(\boldsymbol{x} \mid y = +1), \boldsymbol{x}'_1, \ldots, \boldsymbol{x}'_{n'} \sim p(\boldsymbol{x})\}.$$

Let $\ell_\eta(z)$ be a *surrogate loss* for the zero-one loss

$$\ell_\eta(z) = \begin{cases} 0 & \text{if } z > \eta, \\ 1 - z/\eta & \text{if } 0 < z \le \eta, \\ 1 & \text{if } z \le 0. \end{cases}$$

For any $\eta > 0$, $\ell_\eta(z)$ is lower bounded by $\ell_{0\text{-}1}(z)$ and approaches $\ell_{0\text{-}1}(z)$ as $\eta$ approaches zero. Moreover, let

$$\widetilde{\ell}(yf(\boldsymbol{x})) = \frac{2}{y+3} \ell_{0\text{-}1}(yf(\boldsymbol{x})) \quad \text{and} \quad \widetilde{\ell}_\eta(yf(\boldsymbol{x})) = \frac{2}{y+3} \ell_\eta(yf(\boldsymbol{x})).$$

Then we have the following theorems (proofs are provided in Appendix A). Our key idea is to decompose the generalization error as

$$\boldsymbol{E}_{p(\boldsymbol{x},y)}[\ell_{0\text{-}1}(yf(\boldsymbol{x}))] = \pi^* \boldsymbol{E}_{p(\boldsymbol{x}|y=+1)} \left[ \widetilde{\ell}(f(\boldsymbol{x})) \right] + \boldsymbol{E}_{p(\boldsymbol{x},y)} \left[ \widetilde{\ell}(yf(\boldsymbol{x})) \right],$$

where $\pi^* := p(y = 1)$ is the true class prior of the positive class.

**Theorem 1.** *Fix $f \in \mathcal{F}$, then, for any $0 < \delta < 1$, with probability at least $1 - \delta$ over the repeated sampling of $\{\boldsymbol{x}_1, \ldots, \boldsymbol{x}_n\}$ and $\{(\boldsymbol{x}'_1, y'_1), \ldots, (\boldsymbol{x}'_{n'}, y'_{n'})\}$ for evaluating the empirical error,*[1]

$$\boldsymbol{E}_{p(\boldsymbol{x},y)}[\ell_{0\text{-}1}(yf(\boldsymbol{x}))] - \frac{1}{n'} \sum_{j=1}^{n'} \widetilde{\ell}(y'_j f(\boldsymbol{x}'_j)) \le \frac{\pi^*}{n} \sum_{i=1}^{n} \widetilde{\ell}(f(\boldsymbol{x}_i)) + \left( \frac{\pi^*}{2\sqrt{n}} + \frac{1}{\sqrt{n'}} \right) \sqrt{\frac{\ln(2/\delta)}{2}}. \tag{13}$$

**Theorem 2.** *Fix $\eta > 0$, then, for any $0 < \delta < 1$ with probability at least $1 - \delta$ over the repeated sampling of $\{\boldsymbol{x}_1, \ldots, \boldsymbol{x}_n\}$ and $\{(\boldsymbol{x}'_1, y'_1), \ldots, (\boldsymbol{x}'_{n'}, y'_{n'})\}$ for evaluating the empirical error, every $f \in \mathcal{F}$ satisfies*

$$\boldsymbol{E}_{p(\boldsymbol{x},y)}[\ell_{0\text{-}1}(yf(\boldsymbol{x}))] - \frac{1}{n'} \sum_{j=1}^{n'} \widetilde{\ell}_\eta(y'_j f(\boldsymbol{x}'_j)) \le \frac{\pi^*}{n} \sum_{i=1}^{n} \widetilde{\ell}_\eta(f(\boldsymbol{x}_i)) + \left( \frac{\pi^*}{\sqrt{n}} + \frac{2}{\sqrt{n'}} \right) \frac{C_\alpha C_k}{\eta}$$

$$+ \left( \frac{\pi^*}{2\sqrt{n}} + \frac{1}{\sqrt{n'}} \right) \sqrt{\frac{\ln(2/\delta)}{2}}.$$

In both theorems, the generalization error bounds are of order $O(1/\sqrt{n} + 1/\sqrt{n'})$. This order is optimal for PU classification where we have $n$ i.i.d. data from a distribution and $n'$ i.i.d. data from another distribution. The error bounds for fully supervised classification, by assuming these $n + n'$ data are all i.i.d., would be of order $O(1/\sqrt{n + n'})$. However, this assumption is unreasonable for PU classification, and we cannot train fully supervised classifiers using these $n + n'$ samples. Although the orders (and the losses) differ slightly, $O(1/\sqrt{n} + 1/\sqrt{n'})$ for PU classification is no worse than $2\sqrt{2}$ times $O(1/\sqrt{n + n'})$ for fully supervised classification (assuming $n$ and $n'$ are equal). To the best of our knowledge, no previous work has provided such generalization error bounds for PU classification.

Table 1: Misclassification rate (in percent) for PU classification on the USPS dataset. The best, and equivalent by $95\%$ t-test, is indicated in bold.

| $\pi$ | 0.2 | | 0.4 | | 0.6 | | 0.8 | | 0.9 | | 0.95 | |
|---|---|---|---|---|---|---|---|---|---|---|---|---|
| | Ramp | Hinge | Ramp | Hinge | Ramp | Hinge | Ramp | Hinge | Ramp | Hinge | Ramp | Hinge |
| 0 vs 1 | **3.36** | 4.40 | **4.85** | **4.78** | 5.48 | **5.18** | 4.16 | **4.00** | **2.68** | 9.86 | **1.71** | 4.94 |
| 0 vs 2 | **5.15** | 6.20 | **6.96** | 8.67 | **7.22** | 8.79 | **5.90** | 14.60 | **4.12** | 9.92 | **2.80** | 4.94 |
| 0 vs 3 | **3.49** | 5.52 | **4.72** | 8.08 | **5.02** | 8.52 | **4.06** | 16.51 | **2.89** | 9.92 | **2.12** | 4.94 |
| 0 vs 4 | **1.68** | 2.83 | **2.05** | 4.00 | **2.21** | 3.99 | **2.00** | 3.03 | **1.70** | 9.92 | **1.42** | 4.94 |
| 0 vs 5 | **5.21** | 7.42 | **7.22** | 11.16 | **7.46** | 12.04 | **6.16** | 19.78 | **4.36** | 9.92 | **3.21** | 4.94 |
| 0 vs 6 | **11.47** | **11.61** | 19.87 | **19.59** | 22.58 | 22.94 | **15.13** | 19.83 | **8.86** | 9.92 | 5.29 | **4.94** |
| 0 vs 7 | **1.89** | 3.55 | **2.55** | 4.61 | **2.64** | 3.70 | **2.31** | 2.49 | **1.78** | 9.92 | **1.39** | 4.94 |
| 0 vs 8 | **3.98** | 5.09 | **4.81** | 7.00 | **4.75** | 6.85 | **3.74** | 11.34 | **2.79** | 9.92 | **2.11** | 4.94 |
| 0 vs 9 | **1.22** | 2.76 | **1.60** | 3.86 | **1.73** | 3.56 | **1.61** | 2.24 | **1.38** | 9.92 | **1.13** | 4.94 |

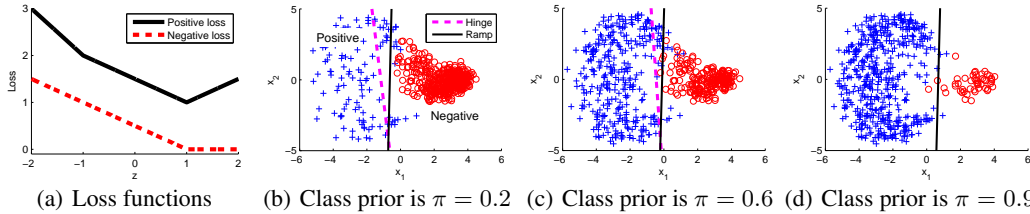

(a) Loss functions    (b) Class prior is $\pi = 0.2$   (c) Class prior is $\pi = 0.6$   (d) Class prior is $\pi = 0.9$.

Figure 4: Examples of the classification boundary for the "0" vs. "7" digits, obtained by PU learning. The unlabeled dataset and the underlying (latent) class labels are given. Since discriminant function for the hinge loss case is constant $1$ when $\pi = 0.9$, no decision boundary can be drawn and all negative samples are misclassified.

## 6 Experiments

In this section, the experimentally compare the performance of the ramp loss and the hinge loss in PU classification (weighting was performed w.r.t. the true class prior and the ramp loss was optimized with [12]). We used the USPS dataset, with the dimensionality reduced to 2 via principal component analysis to enable illustration. $550$ samples were used for the positive and mixture datasets. From the results in Table 1, it is clear that the ramp loss gives a much higher classification accuracy than the hinge loss, especially for large class priors. This is due to the fact that the effect of the superfluous penalty term in (10) becomes larger since it scales with $\pi$.

When the class prior is large, the classification accuracy for the hinge loss is often close to $1 - \pi$. This can be explained by (10): collecting the terms for the positive expectation, we get an effective loss function for the positive samples (illustrated in Figure 4(a)). When $\pi$ is large enough, the positive loss is minimized, giving a constant $1$. The misclassification rate becomes $1 - \pi$ since it is a combination of the false negative rate and the false positive rate according to the class prior.

Examples of the discrimination boundary for digits "0" vs. "7" are given in Figure 4. When the class-prior is low (Figure 4(b) and Figure 4(c)) the misclassification rate of the hinge loss is slightly higher. For large class-priors (Figure 4(d)), the hinge loss causes all samples to be classified as positive (inspection showed that $\boldsymbol{w} = \boldsymbol{0}$ and $b = 1$).

## 7 Conclusion

In this paper we discussed the problem of learning a classifier from positive and unlabeled data. We showed that PU learning can be solved using a cost-sensitive classifier if the class prior of the unlabeled dataset is known. We showed, however, that a non-convex loss must be used in order to prevent a superfluous penalty term in the objective function.

In practice, the class prior is unknown and estimated from data. We showed that the excess risk is actually controlled by an effective class prior which depends on both the estimated class prior and the true class prior. Finally, generalization error bounds for the problem were provided.

**Acknowledgments**
MCdP is supported by the JST CREST program, GN was supported by the 973 Program No. 2014CB340505 and MS is supported by KAKENHI 23120004.

## Footnotes

[1]The empirical error that we cannot evaluate in practice is in the left-hand side of (13), and the empirical error and confidence terms that we can evaluate in practice are in the right-hand side of (13).

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
