[Supplementary Material]

# A  Proofs of Theorems

In this section, we give proofs of theorems.

## A.1  Decomposition of generalization error in PU classification

Assume that $\pi^* := p(y = 1)$ is the true class prior of the positive class. Subsequently,

$$
\begin{aligned}
\boldsymbol{E}_{p(\boldsymbol{x},y)}[\ell_{\text{0-1}}(yf(\boldsymbol{x}))] &= \int_{\boldsymbol{R}^d} \sum_y \ell_{\text{0-1}}(yf(\boldsymbol{x}))p(\boldsymbol{x},y)\mathrm{d}\boldsymbol{x} \\
&= \int_{\boldsymbol{R}^d} \sum_y \widetilde{\ell}(yf(\boldsymbol{x}))\left(\frac{y+3}{2}\right)p(\boldsymbol{x},y)\mathrm{d}\boldsymbol{x} \\
&= \int_{\boldsymbol{R}^d} \sum_y \widetilde{\ell}(yf(\boldsymbol{x}))(2p(\boldsymbol{x},y=+1)+p(\boldsymbol{x},y=-1))\mathrm{d}\boldsymbol{x} \\
&= \pi^* \int_{\boldsymbol{R}^d} \widetilde{\ell}(f(\boldsymbol{x}))p(\boldsymbol{x}\mid y=+1)\mathrm{d}\boldsymbol{x} + \int_{\boldsymbol{R}^d} \sum_y \widetilde{\ell}(yf(\boldsymbol{x}))p(\boldsymbol{x},y)\mathrm{d}\boldsymbol{x} \\
&= \pi^* \boldsymbol{E}_{p(\boldsymbol{x}|y=+1)}\left[\widetilde{\ell}(f(\boldsymbol{x}))\right] + \boldsymbol{E}_{p(\boldsymbol{x},y)}\left[\widetilde{\ell}(yf(\boldsymbol{x}))\right]. \quad (14)
\end{aligned}
$$

This decomposition is the key idea of our error bounds.

## A.2  Proof of Theorem 1

Note that $\widetilde{\ell}$ maps to $[0,1]$, but if $y = +1$ it maps to $[0, 1/2]$. We apply *McDiarmid's inequality* and obtain

$$
\Pr\left\{\boldsymbol{E}_{p(\boldsymbol{x}|y=+1)}\left[\widetilde{\ell}(f(\boldsymbol{x}))\right] - \frac{1}{n}\sum_{i=1}^n \widetilde{\ell}(f(\boldsymbol{x}_i)) \geq \epsilon\right\} \leq \exp\left(-\frac{2\epsilon^2}{n(1/2n)^2}\right).
$$

Equating the right-hand side of the above inequality to $\delta/2$ gives us that with probability at least $1 - \delta/2$,

$$
\boldsymbol{E}_{p(\boldsymbol{x}|y=+1)}\left[\widetilde{\ell}(f(\boldsymbol{x}))\right] - \frac{1}{n}\sum_{i=1}^n \widetilde{\ell}(f(\boldsymbol{x}_i)) \leq \sqrt{\frac{\ln(2/\delta)}{8n}}.
$$

Apply *McDiarmid's inequality* again and obtain that with probability at least $1 - \delta/2$,

$$
\boldsymbol{E}_{p(\boldsymbol{x},y)}\left[\widetilde{\ell}(yf(\boldsymbol{x}))\right] - \frac{1}{n'}\sum_{j=1}^{n'} \widetilde{\ell}(y_j'f(\boldsymbol{x}_j')) \leq \sqrt{\frac{\ln(2/\delta)}{2n'}}.
$$

Combining these two concentration inequalities and Eq. (14) completes the proof. $\qquad\square$

## A.3  Proof of Theorem 2

**Definition 3** ([15], Definitions 3.1 and 3.2). *Let $\mathcal{F}$ be a class of functions. Let $\boldsymbol{x}_1, \ldots, \boldsymbol{x}_n$ be independent observations drawn according to $p(\boldsymbol{x})$, and $\sigma_1, \ldots, \sigma_n$ be independent uniformly $\{\pm 1\}$-valued random variables, i.e., Rademacher variables. The empirical Rademacher complexity of $\mathcal{F}$ conditioned on $\boldsymbol{x}_1, \ldots, \boldsymbol{x}_n$ is defined by*

$$
\widehat{\mathcal{R}}_n(\mathcal{F}) := \boldsymbol{E}_{\sigma_1,\ldots,\sigma_n}\left\{\sup_{f\in\mathcal{F}} \frac{1}{n}\sum_{i=1}^n \sigma_i f(\boldsymbol{x}_i)\right\},
$$

*and the Rademacher complexity of $\mathcal{F}$ is defined by*

$$
\mathcal{R}_n(\mathcal{F}) := \boldsymbol{E}_{\boldsymbol{x}_1,\ldots,\boldsymbol{x}_n}\left\{\widehat{\mathcal{R}}_n(\mathcal{F})\right\}.
$$

Denote by $\mathcal{R}_n(\mathcal{F})$ the Rademacher complexity w.r.t. $p(\boldsymbol{x} \mid y = +1)$, and $\mathcal{R}'_{n'}(\mathcal{F})$ the Rademacher complexity w.r.t. $p(\boldsymbol{x})$. By Theorem 5.5 of [15] and the condition that $C_k = \sup_{\boldsymbol{x} \in \boldsymbol{R}^d} \sqrt{k(\boldsymbol{x}, \boldsymbol{x})}$, we get

$$
\begin{aligned}
\mathcal{R}_n(\mathcal{F}) &\leq \frac{C_\alpha C_k}{\sqrt{n}}, \\
\mathcal{R}'_{n'}(\mathcal{F}) &\leq \frac{C_\alpha C_k}{\sqrt{n'}}.
\end{aligned}
\tag{15}
$$

Next, we need the following lemmas.

**Lemma 4.** *Fix $\eta > 0$, then, for any $0 < \delta < 1$ with probability at least $1 - \delta$ over the repeated sampling of $\{(\boldsymbol{x}'_1, y'_1), \ldots, (\boldsymbol{x}'_{n'}, y'_{n'})\}$ for evaluating the empirical error, every $f \in \mathcal{F}$ satisfies*

$$
\boldsymbol{E}_{p(\boldsymbol{x},y)}\left[\widetilde{\ell}(yf(\boldsymbol{x}))\right] - \frac{1}{n'}\sum_{j=1}^{n'}\widetilde{\ell}_\eta(y'_j f(\boldsymbol{x}'_j)) \leq \frac{2}{\eta}\mathcal{R}'_{n'}(\mathcal{F}) + \sqrt{\frac{\ln(1/\delta)}{2n}}.
$$

*Proof.* Note that both $\widetilde{\ell}$ and $\widetilde{\ell}_\eta$ map to $[0,1]$, $\widetilde{\ell}$ is lower bounded by $\widetilde{\ell}_\eta$, and the Lipschitz constant of $\widetilde{\ell}_\eta$ is $1/\eta$. Hence, this lemma is essentially same as the first half of Theorem 4.4 in [15]. $\square$

**Lemma 5.** *Fix $\eta > 0$, then, for any $0 < \delta < 1$ with probability at least $1 - \delta$ over the repeated sampling of $\{\boldsymbol{x}_1, \ldots, \boldsymbol{x}_n\}$ for evaluating the empirical error, every $f \in \mathcal{F}$ satisfies*

$$
\boldsymbol{E}_{p(\boldsymbol{x}|y=+1)}\left[\widetilde{\ell}(f(\boldsymbol{x}))\right] - \frac{1}{n}\sum_{i=1}^{n}\widetilde{\ell}_\eta(f(\boldsymbol{x}_i)) \leq \frac{1}{\eta}\mathcal{R}_n(\mathcal{F}) + \sqrt{\frac{\ln(1/\delta)}{8n}}.
$$

*Proof.* If we fix $y = +1$, both $\widetilde{\ell}$ and $\widetilde{\ell}_\eta$ map to $[0, 1/2]$, and the Lipschitz constant of $\widetilde{\ell}_\eta$ is $1/(2\eta)$. Then, the proof of this lemma is analogous with the proof of the first half of Theorem 4.4 in [15], while there are two difference points:

- When applying Theorem 3.1 of [15], note that both $\widetilde{\ell}$ and $\widetilde{\ell}_\eta$ map to $[0, 1/2]$, and consequently McDiarmid's inequality results in a tighter bound;

- When applying Lemma 4.2 of [15], note that $\widetilde{\ell}_\eta$ is $(1/(2\eta))$-Lipschitz continuous, and thus the contraction of Rademacher averages results in a tighter bound. $\square$

By Lemma 5 and (15), with probability at least $1 - \delta/2$ over the repeated sampling of $\{\boldsymbol{x}_1, \ldots, \boldsymbol{x}_n\}$,

$$
\boldsymbol{E}_{p(\boldsymbol{x}|y=+1)}\left[\widetilde{\ell}(f(\boldsymbol{x}))\right] - \frac{1}{n}\sum_{i=1}^{n}\widetilde{\ell}_\eta(f(\boldsymbol{x}_i)) \leq \frac{C_\alpha C_k}{\eta\sqrt{n}} + \sqrt{\frac{\ln(2/\delta)}{8n}}.
$$

Similarly, by Lemma 4 and (15), with probability at least $1 - \delta/2$ over the repeated sampling of $\{(\boldsymbol{x}'_1, y'_1), \ldots, (\boldsymbol{x}'_{n'}, y'_{n'})\}$,

$$
\boldsymbol{E}_{p(\boldsymbol{x},y)}\left[\widetilde{\ell}(yf(\boldsymbol{x}))\right] - \frac{1}{n'}\sum_{j=1}^{n'}\widetilde{\ell}_\eta(y'_j f(\boldsymbol{x}'_j)) \leq \frac{2C_\alpha C_k}{\eta\sqrt{n'}} + \sqrt{\frac{\ln(2/\delta)}{2n'}}.
$$

Combining these two concentration inequalities and Eq. (14) completes the proof. $\square$