[Reviews · NeurIPS 2014]

Submitted by Assigned_Reviewer_7

This paper provides some theoretical analysis on Positive and Unlabeled Data learning (PU), when only the positive instances and unlabeled instances are available. The authors show that learning from PU is equivalent to a cost-sensitive classification task if the label prior is known. The main contribution of the paper is that, the authors show that using any convex loss leads to inconsistent classifier for PU tasks. Instead, using a non-convex ramp loss gives a consistent estimator. This theoretical justification is supported by experiments, which demonstrate that adopting hinge loss of SVM may result in very bad classification error comparing to using the non-convex ramp loss. When a non-convex loss is used and can be optimized, the authors have analyzed the generalization error of PU learning and given tight bound.

Overall, the analysis in the paper is interesting and it provides some useful insight. For example, it suggests not to use a hinge loss in the PU setting, even if we know that the data is separable. This is quite different from the theory of classical SVM. On the other hand, the authors have not provided a feasible algorithm for solving the non-convex optimization problem. In section 5, it is claimed that learning in the PU setting can achieve a generalization error no worse than 2\sqrt{2} times the fully supervised case, which is not true. In particular, 1/\sqrt{n}+1/\sqrt{n'} can be arbitrarily worse than 1/\sqrt{n+n'}, when n/n' goes to infinity or when n'/n goes to infinity. In addition, the constants C_alpha and C_k might be sensitively depending on n or n'. Such a dependence should be discussed.
Summary: This is a good paper with interesting analysis and insightful results. Empirical results also support the theoretical conclusion. Although the theory is not complicated, it indeed sheds some light on PU learning, which might be useful in practice.

Submitted by Assigned_Reviewer_18

The paper presents an analysis of learning for a binary classification task when only positive and unlabeled data are available. As the authors point out this setting can be found in many applications, in particular in outlier detection. The authors begin by showing that PU classification can be modelled as a cost-sensitive classification problem. It is also argued that in such settings a non-convex loss is necessary, the argument here is based on the shortcomings of the hinge-loss specifically with respects to the ramp loss. In the modelling proposed by the authors the minimized risk depends on an estimate of the class prior. As in practice one rarely has access to the true prior, the authors introduce the notion of effective prior, which is shown to directly influence the risk. They further show that at least in the cases where the true class prior is large the effective prior is not influenced by small variations in the estimated prior. The authors also provide generalization error bounds in the case of kernel machines. They argue that the expected 0-1 loss in the case of PU classification is only 2*sqrt(2) times that of the fully supervised setting. Finally the paper presents empirical evidence in favour of the ramp loss over the hinge loss.

The paper is overall well written and the subject is I think quite interesting. I am not personally aware of similar work, though I have not worked on PU classification. Multiple facets of the problem are theoretically analyzed, and I think in this respect the paper is not lacking. However there are a number of issues that prevent me from recommending acceptance. I would be interested on hearing from the authors on these points.

- In section 3, I believe the authors overstate their findings. They state (lines 193), that to obtain the correct decision boundary, the loss function should be symmetric and thus the use of a non-convex loss is unavoidable. I don't see how the necessity for symmetry has been proven. In my view all that has been shown is that the ramp loss does not incorporate this "superfluous" term. Furthermore non-convexity does not necessarily follow from symmetry. As an aside, I think it is interesting to note the connection between the hinge loss terms in this settings and the Universum.

- In section 4, line 248 states that the risk when applying \hat{f} is on a line tangent to R* at \pi=\hat{\pi}
and that the difference of \hat{\pi} from true \pi results in the excess risk. However in figure 3, the line is tangent at \pi=\tilde{\pi}. Also, the notation here is not helpful as \pi is used to denote the true prior and a variable.

- In section 5, the authors state that the generalization error bound is at most 2*sqrt(2) times the error bound in the supervised case. I don't see why this holds, it seems to me that the authors are arguing \frac{2\sqrt{2}}{\sqrt{n + n'}} \geq \frac{1}{\sqrt{n'}} + \frac{1}{\sqrt{n}}, but this is not true. In any case, I don't statements of this sort don't make sense when working with big-O classes, as constants are suppressed. Also, in the abstract this relationship is mistakenly stated to relate to the errors, when in fact it relates to the bounds.

- The experimental results are not entirely convincing. To be sure the ramp loss dominates the hinge loss as expected from the analysis. However results are shown only on a single dataset and on a small subset of the binary tasks that can be derived (only those involving zero). I understand that the main contribution is theoretical but it does take away a bit. What is truly missing however is how \hat{\pi} was estimated in these experiments. Was the true prior used?

- A perhaps minor issue, but the notation in section 2 is not clean. P_1 is used for two different distribution P(X|Y=1) and P(f(X)=1|Y=1).
Summary: A well written paper on an interesting subject. There a few issues however that take away from the overall quality of the paper.

Submitted by Assigned_Reviewer_22

The paper analyzes the problem of learning from positive and unlabeled examples (PU learning), which is an important and useful problem to study given it has a lot of applications in multiple domains today. The ideas presented in the paper are relatively simple but the analysis is novel; the theoretical results are non-trivial and worthy of publication.

The paper makes multiple contributions. First it shows that the PU learning problem can be reduced to cost-sensitive classification. Then it proposes the use of ramp loss function in practice as a proxy to minimize 0-1 loss in the PU learning setting. An important practical concern of selecting the class prior is then addressed. Finally, generalization error bounds for PU learning are presented. Overall, the paper is well-written. Below are some concerns and suggestions to improve the paper:

1) The idea of using cost-sensitive classification for PU learning is known before, although under different assumptions. For example, [1] and weighted logistic regression approach (Lee and Liu, ICML 2003). The paper makes no connections to such existing approaches that cast PU learning problem as supervised learning problem with noisy labels (by treating all unlabeled data as negatives, as in this paper). This discussion is essential (in Section 2, perhaps) for elucidating the novelty of the paper.

2) The key property for consistency of PU classification seems to the "symmetry" of the ramp loss. It is known that other (non-convex) loss functions such as Probit and Sigmoid losses satisfy this property too. See "Ghosh, Aritra, Naresh Manwani, and P. S. Sastry, Making Risk Minimization Tolerant to Label Noise, arXiv preprint arXiv:1403.3610 (2014)." Currently, ramp loss choice seems to be ad hoc. Rewriting Section 3 with the focus on symmetry property and giving other examples of loss functions would make it stronger.

3) The paper never makes the actual algorithm for PU classification explicit. It would be good to list the algorithm for cost-sensitive classification with class prior estimation clearly.

4) The generalization bounds in Section 5 are given for ERM with respect to a certain surrogate loss (\ell_\eta(z)). This looks similar to ramp loss. The discrepancy between experiments (where ramp loss is used) and the surrogate loss used in Section 5 is not discussed, let alone justified. Also, $\ell_\eta$ doesn't seem to satisfy the symmetry property. Isn't this an issue? These seem to be the weak links in the paper.

Minor comments:
1. $\pi$ is not defined until much later its use in Section 2.
2. The ramp loss definition looks wrong in Figure 1.

I have read the rebuttal and I have no pressing concerns. My decision remains accept.
Summary: The paper casts PU classification as cost-sensitive learning. Though the idea of using cost-sensitive learning when there are no negative examples to train has been known before, the analysis presented in the paper is novel. The paper makes multiple contributions for PU classification, but the discrepancy between analysis and experiments seems to be overlooked by the authors.
Author Feedback
Author rebuttal: Assigned_Reviewer_18

Q: "They state (lines 193), that to obtain the correct decision boundary, the loss function should be symmetric and thus the use of a non-convex loss is unavoidable. I don't see how the necessity for symmetry has been proven. In my view all that has been shown is that the ramp loss does not incorporate this "superfluous" term."

A: The superfluous penalty term may cause the wrong decision boundary to be selected even though the correctly specified decision boundary is in the model. This is the essence of the example in Figure 2.

Q: "Furthermore non-convexity does not necessarily follow from symmetry."

A: The function \ell_R(z) should be non-convex if it is symmetric in the sense of line 161 because \ell_R(z) = 1-\ell_R(-z)

If \ell_R(z) is convex, then \ell_R(-z) is convex. Therefore 1-\ell_R(-z) should be concave. Unless \ell_R(z) is a function that is both convex and concave (e.g. linear or constant function), the equality is a contradiction. This implies that the loss function should be non-convex.

Q: "In section 4, line 248 states that the risk when applying \hat{f}..."

A: The explanation in this section will be clarified. Training is performed with the estimated class prior \hat{\pi}.

In ordinary classification (which is discussed in line 248), the change is with respect to the estimated class prior \hat{\pi}.

In PU learning, the change is with respect to the effective class prior \tilde{\pi}. The effective class prior depends on both the estimated class prior and the unknown true class prior.

Q: "In section 5, the authors state that the generalization error bound is at most 2*sqrt(2) times the error bound in the supervised case."

A: This is a misstatement and will be corrected -- it is 2*sqrt(2) only when n=n'. The discussion on the bounds will be clarified.

Q: "The experimental results are not entirely convincing. To be sure the ramp loss dominates the hinge loss as expected from the analysis. However results are shown only on a single dataset and on a small subset... What is truly missing however is how \hat{\pi} was estimated in these experiments. Was the true prior used?"

A: The goal of the experiments was to show the failure of the hinge loss in the PU learning framework. For this reason the MNIST dataset was chosen. The results for other combinations of digits will be added in the appendix.

The true class-prior was used in this experiment since the goal is to show the failure of the hinge loss. Experiments will be added to the appendix to show the effect of a change in class prior.

Assigned_Reviewer_22

1) The current paper approaches the problem from a loss function perspective. Using such loss function, the discriminant boundary may be directly estimated. The papers [1] and Lee and Liu (ICML 2003), use logistic regression as intermediate steps. We will add a discussion and contrast these works with our own in Section 2.

2) We thank the reviewer for bringing the Ghosh et al. (2014) paper to our attention. The hinge loss was chosen since it is used by SVM. The ramp-loss is a natural non-convex counterpart. Another motivation is that there are also research and implementations for solving the non-convex ramp-loss problem (e.g., CCCP of [12] or [13]).

3) For the ramp-loss, the classifier minimizes the empirical version with a regularizer. An implementation of the CCCP algorithm in [12] was used in the ramp-loss case. The implementation will be discussed in the appendix.

4) Please allow us to clarify this point. There are two types of loss functions in our manuscript, one for evaluating the classification error and one for training the classification model. The 0-1 loss in the bound and experiments and its surrogate in the bound are the first type, while the hinge loss and the ramp loss elsewhere are the second
type. These two types are used for different purposes and thus share different properties. Therefore, using the "non-symmetric" surrogate loss in the bound is not an issue.

Minor comments (1) and (2) will be fixed.

Assigned_Reviewer_7

Q: "On the other hand, the authors have not provided a feasible algorithm for solving the non-convex optimization problem"

A: In all experiments an implementation of the CCCP algorithm using the ramp loss of [12] was used. Since this aspect does not form the authors' contribution, it was omitted from the paper. We will provide a discussion of the CCCP algorithm of [12] in the appendix.

Q: "In section 5, it is claimed that learning in the PU setting can achieve a generalization error no worse than 2\sqrt{2}"

A: This was not clearly stated in the paper. The 2\sqrt{2} is only true when n=n'. We will clarify that discussion and provide a further discussion as the reviewer suggests.